

# Temperament and character profiles of medical students associated with tolerance of ambiguity and perfectionism

Janni Leung[1], C. Robert Cloninger[2], Barry A. Hong[2], Kevin M. Cloninger[3] and Diann S. Eley[1]

[1] The University of Queensland, Brisbane, Australia
[2] Department of Psychiatry, School of Medicine, Washington University, St. Louis, United States of America
[3] The Anthropedia Foundation, St. Louis, United States of America

## ABSTRACT

**Background**. Certain personal attributes, such as perfectionism and tolerance of ambiguity, have been identified as influential in high achieving students. Medical students have been identified as high achievers and perfectionistic, and as such may be challenged by ambiguity. Medical students undertake a long and challenging degree. Personality has been shown to influence the well-being and coping and may equip some students to better cope with challenges. This paper examines the association between temperament and character personality profiles with measures of tolerance of ambiguity and with both adaptive and maladaptive constructs of perfectionism.

**Methods**. A self-report questionnaire collected data on a sample of 808 Australian medical students in 2014 and 2015. Personality was measured using the Temperament and Character Inventory (TCIR-140) and classified traits as profiles using a latent class analysis. Two profiles were found. Profile 1 was characterized by low-average levels of Harm Avoidance, and high to very high levels of Persistence, Self-Directedness and Cooperativeness. Moderately-high levels of Harm Avoidance and high levels of Persistence, Self-Directedness and Cooperativeness characterized Profile 2. Moderation regression analyses were conducted to examine the association between the personality profiles with levels of Tolerance of Ambiguity (MSAT-II), Perfectionism-Concern over Mistakes and Perfectionism-High Standards (FMPS), considering demographic characteristics.

**Results**. Students with Profile 1 were higher in levels of Tolerance of Ambiguity, and Perfectionism-High Standards, and lower levels of Perfectionism-Concern over Mistakes compared to Profile 2. These findings remained statistically significant after adjusting for age and gender. A significant personality by age interaction on Tolerance of Ambiguity was found. While higher levels of Tolerance of Ambiguity were associated with older age overall, it remained low across age for students with a personality Profile 2.

**Conclusions**. A particular combination of personality traits was identified to be associated with low Tolerance of Ambiguity and high levels of maladaptive Perfectionism. An intolerance of ambiguity and over concern about mistakes may be maladaptive and underlie vulnerability to stress and poor coping. The psychobiological model of personality provides insight into traits that are stable and those that can be self-regulated through education and training. The interaction between biological mechanisms and socio-cultural learning is relevant to a sample of medical students because it accounts

Corresponding author
Janni Leung, j.leung1@uq.edu.au

for interaction of the biological or innate aspects of their personal development within an intense and competitive learning environment of medical school.

# INTRODUCTION

The health and well-being of medical students along with their successful progression through medical training is an important concern to medical educators. High levels of stress, depression and anxiety are increasingly noted in doctors and medical students (*Firth-Cozens, 2003*; *Dyrbye & Shanafelt, 2006*), and are higher than their nonmedical peers in the general population (*Dyrbye et al., 2011*). In turn, this higher level of dysphoria may affect expressions of empathy, impair professional behaviour and contribute to burn-out and drop-out (*Brazeau et al., 2010*). The many pressures and demands associated with medical training can drive students to excessive self-criticism, self-doubt and fear about making mistakes (*Firth-Cozens, 2003*; *McManus, Keeling & Paice, 2004*; *Rogers, Creed & Searle, 2012*).

Comprehensive reviews have recognised personality as an influencing factor on many aspects of medical students' successful and healthy progression through medical training (*Doherty & Nugent, 2011*; *Hojat, Erdman & Gonnella, 2013*). The literature is clear that certain personality traits can influence or predict various aspects of academic performance, (*Hojat et al., 2003*; *Haight, Chibnall & Schindler, 2012*; *Lievens et al., 2002*; *Ferguson, James & Madeley, 2002*; *McManus & Powis, 2007*), including subsequent performance in postgraduate training (*Hodgson et al., 2007*), subclinical competence, (*Ferguson et al., 2003*; *Hojat, Callahan & Gonnella, 2004*) expression of empathy, (*Hojat et al., 2013*; *Hojat et al., 2014*) subjective well-being, (*Diener & Lucas, 1999*; *Haslam, Whelan & Bastian, 2009*) mental toughness (*Horsburgh, Schermer & Veselka, 2009*), attitudes to work, burnout and stress (*McManus, Keeling & Paice, 2004*), and career interests (*Borges & Osmon, 2001*; *Hojat & Zuckerman, 2008*; *Duffy, Borges & Harthung, 2009*).

The measurement of personality across these studies differ, primarily around variants of the Five-Factor Model (FFM). (*Costa & McCrae, 1992*; *Tyssen et al., 2007*; *Knights & Kennedy, 2007*) The Temperament and Character Inventory (TCI), developed by Cloninger (*Cloninger, Svrakic & Pryzbeck, 1993*; *Cloninger, Svrakic & Wetzel, 1994*), distinguishes between seven personality traits within the domains of temperament (Novelty Seeking, Harm Avoidance, Reward Dependence, Persistence), and character (Self-Directedness, Cooperativeness, Self-Transcendence). Japanese studies with medical students demonstrated positive associations between intrinsic academic motivation and the TCI's Persistence, Self-Directedness and Self-Transcendence, leading to better academic performance (*Tanaka et al., 2009*), and Jiang (*Jiang et al., 2003*) found positive associations between the TCI's Harm Avoidance and measures of anxiety. Resilience was shown to be

strongly correlated with high levels of Self-Directedness, Cooperativeness and Persistence and low Harm Avoidance in Australian medical residents (*Eley et al., 2013*). Furthermore, several studies in the general population have shown that high Self-Directedness and low Harm Avoidance are the strongest predictors of well-being and satisfaction with life (*Cloninger & Zohar, 2011*; *Cloninger, Salloum & Mezzich, 2012*; *Sievert et al., 2016*; *Grucza & Goldberg, 2007*).

Importantly, the TCI provides insight into the combination of traits from a psychobiological perspective identifying aspects of personality that are relatively stable (temperament traits), and those that are more amiable to change through sociocultural learning (character traits). The psychobiological model of personality suggests there is interaction between biological mechanisms and learning over the development of an individual within the context of his/her environment. Heritable influences that refer to genetically inherited qualities are equally great on temperament and character but character plays the self-regulatory role, which is targeted in parenting, teaching, and coaching. Examining profiles or combinations of trait levels affords a more meaningful understanding of what drives individuals' behaviours and how they adapt to life situations (*Tyssen et al., 2007*). The individual traits of a person must interact in a complex dynamic way to adapt to internal or external challenges (*Cloninger & Zohar, 2011*; *Cloninger, Salloum & Mezzich, 2012*). This suggests that certain personality profiles are prone to being more or less vulnerable to adapting to and coping with these challenges. In addition to temperament and character, many other psychological attributes can influence one's capacity for coping and may influence well-being. Pertinent to medical students are the attributes of ambiguity tolerance and perfectionism (*Howe, Smajdor & Stockl, 2005*; *Geller, 2013*; *Enns, Cox & Sareen, 2001*). We chose these attributes as proxy measures of capacity for coping and are discussed briefly below.

The early work by Budner showed that individuals with a low Tolerance of Ambiguity experience anxiety or stress when they perceive new or complex situations (*Budner, 1962*). Geller provides a thoughtful perspective on Tolerance of Ambiguity among medical students (*Geller, 2013*). She emphasises the ubiquitous place of ambiguity in medical practice and the need to understand the ramifications at both ends of the tolerance spectrum. Ambiguity tolerance exerts a strong influence on the attitudes and behaviours of medical students. In particular, low Tolerance of Ambiguity has been linked to higher perceived stress (*Caulfield et al., 2014*) and higher rates of burnout (*Cooke, Doust & Steele, 2013*).

Socio-demographic characteristics such as sex and age are associated with individual differences in ambiguity tolerance. Higher levels are found in men and older students (*Caulfield et al., 2014*), in leaders (*Sherrill, 2001*), and in people willing to work in rural or underserved areas (*Wayne et al., 2011*). In contrast, low ambiguity tolerance has been associated with negative attitudes to underserved groups of people (*Wayne et al., 2011*), burnout in primary care (*Bachman & Freeborn, 1999*) and emergency physicians (*Kuhn, Goldberg & Compton, 2009*), higher levels of stress in medical students (*Caulfield et al., 2014*), and a fear of making mistakes (*West et al., 2009*). Personality traits have also been associated with the way physicians deal with uncertainty

and their diagnostic decision making (*Schneider, Linde & Buhner, 2014*) , but little research has investigated the relationship between temperament and character traits with Tolerance of Ambiguity in medical students.

Perfectionism is a common characteristic of high achieving individuals and is prevalent in medical students and physicians (*Feinmann, 2011*; *Peters & King, 2012*). Perfectionism most commonly features the setting of excessively high personal standards of performance (*Flett & Hewitt, 2002*) which may have important benefits to medical care such as attention to detail and a strong sense of responsibility (*Peters & King, 2012*). Perfectionism exists on a continuum and can be either healthy and adaptive or self-defeating and maladaptive (*Enns, Cox & Sareen, 2001*; *Frost et al., 1990*). The major distinction is that adaptive perfectionists have a higher acceptance to allow minor flaws in their work and still accept it as successful. They are driven by meaningful goals and striving for achievement. In contrast, maladaptive perfectionists are simply unable to accept mistakes. They set standards so unrealistically high that instead of directing their goals toward striving for achievement, they are driven by fear of failure which has implications for decision making. Research by Enns (*Enns, Cox & Sareen, 2001*) found that medical students differed from arts students in aspects of perfectionism. In medical students, adaptive perfectionism was correlated with academic performance and levels of conscientiousness, whereas maladaptive perfectionism correlated with symptoms of distress and levels of neuroticism which further predicted depression over time. Similarly, the association between personality profiles and the different types of perfectionism in medical students has not previously between examined.

The aim of this paper was to examine the relationship between personality profiles of medical students and psychological attributes that may be indicative of coping with a challenging medical degree. We examined the association between personality profiles with tolerance of ambiguity and the two constructs of Perfectionism, (High Standards and Concern over Mistakes), and whether the relationships differ by demographic characteristics. We hypothesised that there would be significant relationships between medical student's personality profile and levels of each coping attribute. We expected that students who fit in a personality profile that comprised a temperament low in Harm Avoidance and high in Persistence, and a character high in Self Directedness and Cooperativeness, would have high levels of ambiguity tolerance and adaptive perfectionism.

## MATERIALS & METHODS

### Design

The study design is quantitative cross-sectional using a self-report questionnaire. Ethics was approved by the Behavioural and Social Sciences Ethical Review Committee at The University of Queensland. (Approval Number: 2015001895). Participants provided written consent which was documented on the questionnaire and approved by this ethics committee.

## Participants and setting

Data were collected in 2014 and 2015 at The University of Queensland School of Medicine, Australia. The questionnaire was identical in both years and was accessed via an online link (Survey Monkey). Students were invited to complete the questionnaire via an email invitation to access the link. The 2014 collection invited all students in the four-year medical program to participate through an invitation to access the link on the student community website. However, this method failed to reliably reach all students. In 2015 the data collection involved only Year 1 students who were invited to participate during a regularly scheduled activity. All participants were graduate entry medical students and completed the survey only once. The overall response rate was 62%. The final sample size was $N = 808$. The mean age was 24.88 years (SD = 4.00), median age was 24 (IQR = 5). The modal age was 23, but the distribution was skewed to the right (skewness = 2.87), due to the mature age entry students. Due to the distribution, age was not able to be analyzed as a continuous variable. Therefore, based on the mean value, the age variable was split into under 25 years, and 25 year or older because this is largely in line with the phases of training in the medical degree.

## Measures

### Temperament and character personality profiles

Personality profiles were measured using the 140-item Temperament and Character Inventory (*Cloninger, Svrakic & Pryzbeck, 1993*; *Cloninger, Svrakic & Wetzel, 1994*). The four temperament traits measure basic emotional drives, are heritable and do not change significantly over the life course. Novelty Seeking reflects a heritable bias observed as exploratory activity in response to novelty, impulsiveness, and extravagance in approach to cues of reward. Harm Avoidance involves a heritable bias observed as anxiety proneness, and pessimistic worry in anticipation of problems. Reward Dependence is behaviour in response to cues of social reward and is observed as social sensitivity and dependence on approval by others. Persistence is the maintenance of behaviour despite frustration, fatigue and reinforcement. It reflects industriousness and determination. The three character traits are developmental and influenced by learning, the environment and life experiences. Self-Directedness measures intra-personal character strengths, such as being responsible, purposeful, goal-oriented and self-confident. Cooperativeness measures inter-personal character strengths, such as tolerance, empathy, and respectful acceptance of the opinions and behaviours of others. Self-transcendence measures transpersonal character strengths. It quantifies the extent to which individuals conceive themselves as integral parts of the universe as a whole.

The TCIR-140 uses a five point Likert scale (1 = absolutely false to 5 = absolutely true). Internal reliability (Cronbach alphas) of each trait ranged from 0.86 to 0.89 for the character and 0.69 to 0.91 for temperament scales. The scales are multifaceted with high and low descriptors of each trait see (*Eley et al., 2016*). Details on the interpretation of low and high scores of the seven TCI subscales are available in Appendix A.

From the TCI subscales, two personality profiles were classified based on clusters of individuals with distinctive combinations of the seven traits revealed in a latent profile
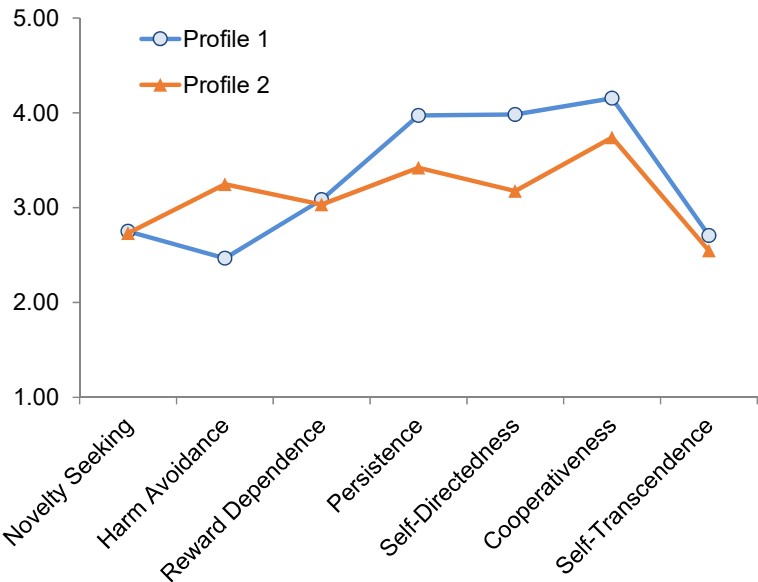

**Figure 1** **The two personality profiles classified based on the seven TCI subscales (N = 808).**

analysis. Further details of the classification methods are available in Appendix B. Profile 1 was characterized by low to average levels of Harm Avoidance, and high to very high levels of Persistence, Self-Directedness and Cooperativeness (see Fig. 1), In comparison, Profile 2 was characterised by average-high levels of Harm Avoidance and high levels of Persistence, Self-Directedness and Cooperativeness, as determined by comparison to normal range percentiles of the population (*Josefsson et al., 2011*). In other words, medical students in both profiles describe themselves as healthier and more mature than average. However, the medical students in Profile 1 are even more confident and resilient than the other medical students and the general population, whereas those in Profile 2 are slightly more anxious and worried than most people in the general population.

## Tolerance of ambiguity

The Multiple Stimulus Types Ambiguity Tolerance Scale-II (MSTAT-II) (*Mclain, 2009*) is based on the theory that an individual's perception of ambiguity is an orientation ranging from attraction to aversion toward stimuli that are uncertain, insoluble or unfamiliar. The MSTAT-II consists of 13 items and uses a Likert response scale of 1 = definitely false to 5 = definitely true. Items were summed, with a low score indicating a low, and high score, a high Tolerance of Ambiguity, alpha = 0.82.

## Perfectionism

Two dimensions of the Frost Multi-dimensional Perfectionism Scale (FMPS) (*Frost et al., 1990*; *Stallman, 2011*) were used; Concern over Mistakes (8 items; maladaptive) is central to the concept of perfectionism and a major component of other measures which tend toward psychological distress including procrastination and fear of mistakes, and High Standards (5 items; adaptive) which reflects positive and healthy striving toward goals. Each item

was presented as a five point Likert scale from 1 = strongly disagree to 5 = strongly agree. Items were summed to derive scores for the two subscales, with higher scores indicative of higher levels of perfectionism. The alphas for each are 0.89 and 0.80 respectively.

### Analysis

Independent samples $t$-tests were conducted, with eta ($\eta$) for assessing effect sizes, comparing the means of the coping attributes (Tolerance of Ambiguity, perfectionism high standards and concern over mistakes), by the two personality profiles. Generalized linear models were used to examine the main effects of personality profile on the coping attributes after including age and gender in the analysis. The coping attributes were the dependent variables in the models. In addition, interaction terms between personality profile with age and gender were included to examine if the association between profiles and the coping attributes differ by demographic characteristics. All analyses were conducted using SPSS 24.

## RESULTS

### Sample characteristics

Descriptive statistics of the participants are presented in Table 1. Slightly over half were under 25 years of age, 54.3% were male. These proportions are not different to the cohorts from which our sample was taken. Over half of the students fit into personality Profile 1 (59.8%).

The 25 or older age group had slightly higher level of Tolerance of ambiguity and perfectionism high standards than the under 25 years group, though the effect sizes were very small (see Table 2). Males had slightly higher levels of Tolerance of Ambiguity but lower levels of perfectionism concern over mistakes than female, though the effect sizes were very small. There were no significant differences by Year of medical school, therefore it was not included in further analyses.

### Coping attributes by personality profiles

Independent t-tests results showed that students with the Profile 1 had significantly higher levels of Tolerance of Ambiguity ($t = 12.73$, $\eta = .58$, $p < .001$), perfectionism high standards ($t = 6.02$, $\eta = .21$, $p < .001$), and lower levels of Perfectionism-Concern over Mistakes ($t = -11.19$, $\eta = .37$, $p < .001$; see Fig. 2).

Results from the generalized linear models showed that personality profiles remained significantly associated with Tolerance of Ambiguity, Perfectionism-Concern over Mistakes, and Perfectionism-High Standards, after taking into consideration the age and gender of the students (see Table 3). A significant Profile by age interaction was found on Tolerance of Ambiguity. The association between profile and Tolerance of Ambiguity was stronger in the older age group. That is, the positive effects of having personality Profile 1 on higher levels of Tolerance of Ambiguity was especially significant for the students aged 25 years or older (see Fig. 3).

**Table 1** Descriptive statistics of the whole sample ($N = 808$).

| | n | % |
|---|---|---|
| **Age** | | |
| Under 25 years | 443 | 54.8 |
| 25 or older | 349 | 43.2 |
| **Gender** | | |
| Male | 439 | 54.3 |
| Female | 368 | 45.5 |
| **Year medical school** | | |
| Year 1 | 599 | 74.1 |
| Year 2 | 75 | 9.3 |
| Year 3 | 67 | 8.3 |
| Year 4 | 66 | 8.2 |
| **TCI personality profile**[a] | | |
| Profile 1 | 483 | 59.8 |
| Profile 2 | 325 | 40.2 |
| **Coping mechanisms** | **Mean** | **SD** |
| Tolerance of Ambiguity | 43.55 | 7.58 |
| Maladaptive perfectionism (concern over mistakes) | 20.63 | 6.14 |
| Adaptive perfectionism (high standards) | 18.46 | 3.84 |

Notes.

[a]Profile 1 was characterized by low to average levels of Harm Avoidance, and high to very high levels of Persistence, Self-Directedness and Cooperativeness. Profile 2 was characterized by average-high levels of Harm Avoidance and high levels of Persistence, Self-Directedness and Cooperativeness.

**Table 2** Mean comparisons of Tolerance of ambiguity and Perfectionism by age, gender and year of study ($N = 808$).

| | Tolerance of Ambiguity | | Perfectionism - Concern over mistakes | | Perfectionism - High Standards | |
|---|---|---|---|---|---|---|
| | **Mean** | **SD** | **Mean** | **SD** | **Mean** | **SD** |
| **Age** | | | | | | |
| Under 25 years | 42.85 | 7.14 | 20.50 | 6.02 | 18.14 | 3.93 |
| 25 or older | 44.42 | 8.09 | 20.78 | 6.36 | 18.93 | 3.67 |
| | $t = 2.90, p = 0.004$ eta-sq $= 0.01$ | | $t = 0.63, p = 0.530$ eta-sq $= 0.00$ | | $t = 2.89, p = 0.004$ eta-sq $= 0.01$ | |
| **Gender** | | | | | | |
| Male | 44.23 | 7.40 | 20.08 | 6.01 | 18.26 | 3.90 |
| Female | 42.74 | 7.73 | 21.28 | 6.25 | 18.71 | 3.77 |
| | $t = 2.78, p = 0.006$ eta-sq $= 0.01$ | | $t = 2.77, p = 0.006$ eta-sq $= 0.01$ | | $t = 1.64, p = 0.101$ eta-sq $= 0.00$ | |
| **Year medical school** | | | | | | |
| Year 1 | 43.82 | 7.64 | 20.38 | 5.92 | 18.54 | 3.93 |
| Year 2 | 43.67 | 7.35 | 21.10 | 6.96 | 18.04 | 3.36 |
| Year 3 | 42.04 | 7.15 | 20.81 | 5.92 | 18.43 | 3.84 |
| Year 4 | 42.14 | 7.25 | 22.23 | 7.24 | 18.29 | 3.62 |
| | $F = 1.93, p = 0.123$ eta-sq $= 0.01$ | | $F = 1.99, p = 0.114$ eta-sq $= 0.01$ | | $F = 0.43, p = 0.729$ eta-sq $= 0.00$ | |

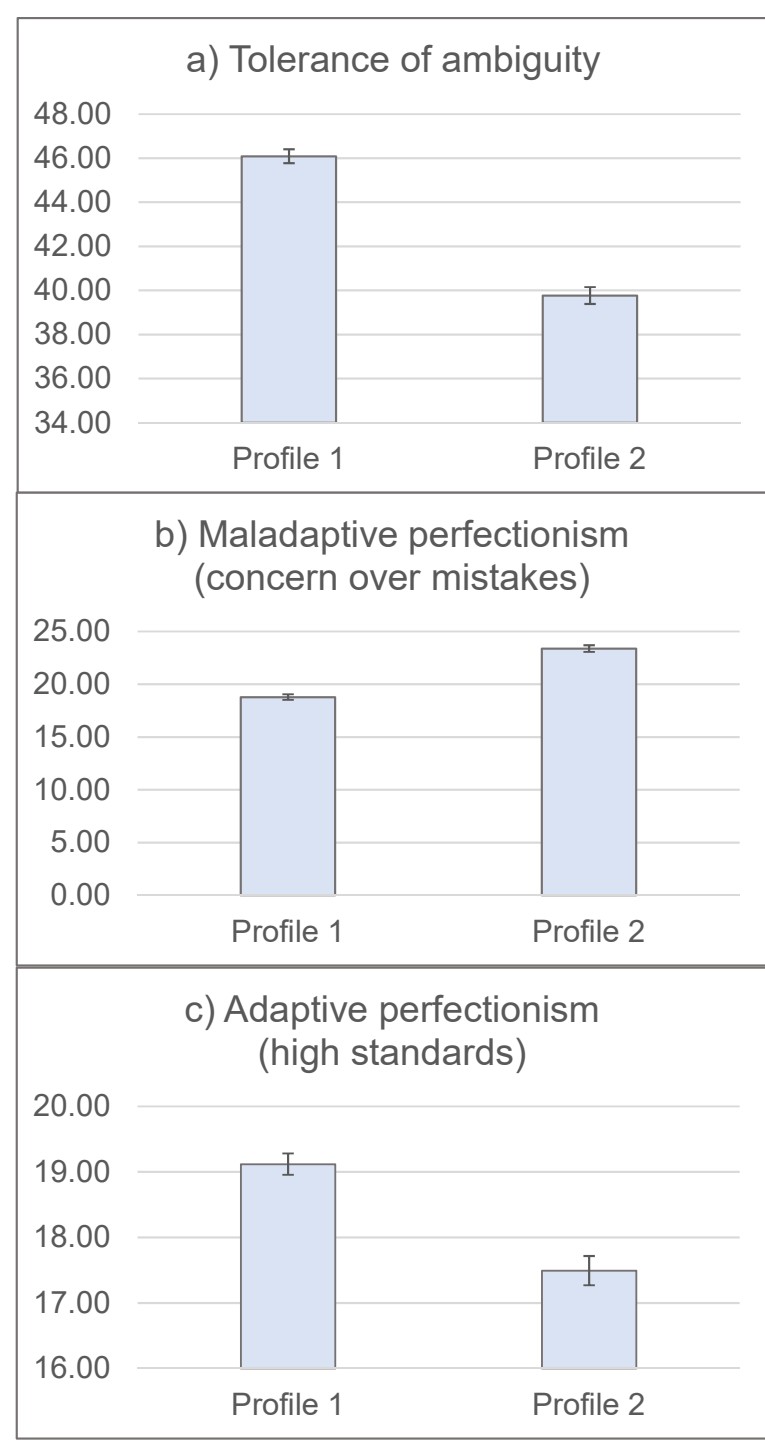

**Figure 2  Levels of ambiguity tolerance and perfectionism by the two personality profiles ($N = 808$).**
Students with the Profile 1 had significantly higher levels of Tolerance of Ambiguity ($t = 12.73$, $\eta = .58$, $p < .001$), Perfectionism-High Standards ($t = 6.02$, $\eta = .21$, $p < .001$), and lower levels of Perfectionism-Concern over Mistakes ($t = -11.19$, $\eta = .37$, $p < .001$).

**Table 3 Generalized linear models with the main effects of personality profiles † and its interaction with age and gender on tolerance of ambiguity and perfectionism.**

|  | B | SE | p |
|---|---|---|---|
| **Model 1: Tolerance of ambiguity** | | | |
| Personality profile | 7.42 | 0.96 | <0.001 |
| Age | −0.21 | 0.78 | 0.789 |
| Gender | 1.26 | 0.77 | 0.103 |
| Personality profile × Age interaction | −2.33 | 1.00 | 0.020 |
| rsonality profile × Gender interaction | 0.58 | 1.00 | 0.564 |
| **Model 2: Maladaptive perfectionism (concern over mistakes)** | | | |
| Personality profile | −5.11 | 0.80 | <0.001 |
| Age | 1.24 | 0.65 | 0.056 |
| Gender | −1.18 | 0.64 | 0.068 |
| Personality profile × Age interaction | 1.12 | 0.84 | 0.182 |
| Personality profile × Gender interaction | −0.34 | 0.83 | 0.681 |
| **Model 3: Adaptive perfectionism (high standards)** | | | |
| Personality profile | 1.10 | 0.52 | 0.035 |
| Age | 1.28 | 0.43 | 0.003 |
| Gender | −0.36 | 0.42 | 0.393 |
| Personality profile × Age interaction | 0.90 | 0.55 | 0.099 |
| Personality profile × Gender interaction | −0.11 | 0.54 | 0.845 |

**Notes.**

Reference groups in the models were personality profile 2, under 25 years age group, and female gender.

† Profile 1 was characterized by low to average levels of Harm Avoidance, and high to very high levels of Persistence, Self-Directedness and Cooperativeness. Profile 2 was characterized by average-high levels of Harm Avoidance and high levels of Persistence, Self-Directedness and Cooperativeness.

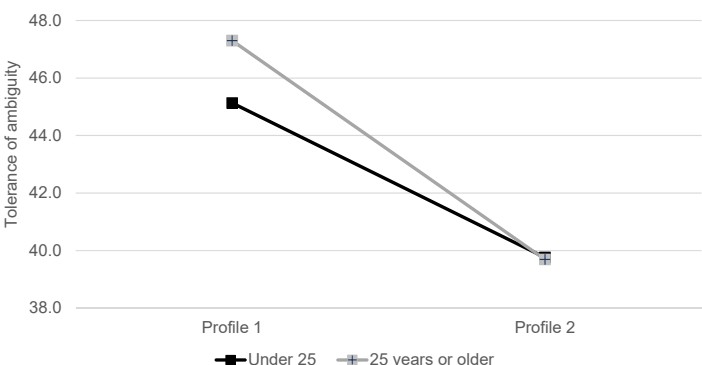

**Figure 3 Interaction effects of personality profile and age on tolerance of ambiguity.**

# DISCUSSION

This study examined Tolerance of Ambiguity, and Perfectionism and their association with temperament and character personality profiles in a sample of medical students. We found that students with the combination of traits identified in Profile 1 (characterized by low to average levels of Harm Avoidance, and high to very high levels of Persistence,

Self-Directedness and Cooperativeness) had significantly higher levels of Tolerance of Ambiguity and lower levels of Perfectionism-Concern over Mistakes compared to those in Profile 2 (characterized by average-high levels of Harm Avoidance and high levels of Persistence, Self-Directedness and Cooperativeness). We identified that students with the particular combinations of personality traits fitting in Profile 2 are implied to be a risk of lower ability to cope in the medical setting. In testing for moderation effects by age, our study highlighted that while Tolerance of Ambiguity may increase with age, it did not do so for students with personality Profile 2, for which Tolerance of Ambiguity remained low across age.

The difference between the Profiles in Perfectionism-High Standards was slight with a small effect size and suggests that medical students in general have high standards and strive for high achievement. This concurs with Enns (*Enns, Cox & Sareen, 2001*) who found medical students to be significantly higher in levels of Perfectionism-High Standards compared to general arts students. Across the whole sample, irrespective of Profile, older students and males had higher levels of Tolerance of Ambiguity (*Caulfield et al., 2014*).

As noted in a previous study (*Eley et al., 2016*), both personality profiles are similar in the trend of their trait levels and it is primarily the levels of two temperament and one character trait that differentiates them. Looking at the two temperament traits; Persistence (higher in Profile 1) is strongly associated with perfectionism and can contribute to maladaptive characteristics (*Cloninger et al., 2012*). This is because of the persistent need for perfection and drive to attain unreasonable standards, self-criticism, and self-doubt is defeatist (*Enns, Cox & Sareen, 2001*). These feelings may lead to maladaptive behaviours such as procrastination or even cheating. However, despite higher levels of Persistence, compared to Profile 2, Profile 1 students still showed lower levels of maladaptive perfectionism (Concern over Mistakes). This effect may be explained by the lower levels of another temperament trait, Harm Avoidance. Low Harm Avoidance enables confidence and optimism along with a degree of comfort with uncertainty. Along with very high Self-Directedness, portrayed as being conscientiousness and goal directed, these two traits act to temper the potential harm of very high Persistence. Rather than a collection of separate parts, these three traits, (low Harm Avoidance and high Self-Directedness and Persistence) demonstrate that multiple dimensions of personality interact as components of a complex adaptive system (*Cloninger & Zohar, 2011*; *Cloninger et al., 2012*) to influence levels of perfectionism (*Enns, Cox & Sareen, 2001*) and ultimately through more effective coping and levels of resilience (*Eley et al., 2013*).

This same combination of temperament and character traits (Profile 1) strengthens our understanding of ambiguity tolerance. To put it in the context of our study, the ubiquitous ambiguity present in medicine (*Geller, 2013*), implies a need for more consideration on how to address low ambiguity tolerance and its possible association with stress, burnout, quality of patient care (*Caulfield et al., 2014*; *Cooke, Doust & Steele, 2013*; *Wayne et al., 2011*), and not surprisingly an over concern for mistakes (*West et al., 2009*).

Our study design only allows us to suggest the utility of the knowing the associations we have found in the data. Nevertheless, we posit that knowing you are low in Tolerance of Ambiguity, or excessively concerned about mistakes may not be as helpful as understanding

why you tend towards those feelings and attitudes. As suggested by Epstein (*Epstein & Krasner, 2013*), understanding one's own personality helps build self-awareness, to recognise when stressors occur, and self-reflection on how to react or self-monitor one's feelings and coping response. Our study was only cross-sectional and exploratory. A longitudinal prospective design would be the only way to determine if a better understanding of one's personality heightens self-awareness and better equips them to perceive and deal with challenges and hardships. It would also enable us to further examine how personality and coping interacts over time as the students age and transition into medical practice. Our data is fed back to students, both individually on request, and in aggregate to help stimulate an interest in reflecting on how they personally cope (or not), with what they find challenging in medical school. Further research will follow students through medical school to examine changes in personality profiles and measures of coping attributes over time.

There are limitations to this study which include the inherent bias in self-report data collection and the cross-sectional design with one medical school, which may reduce generalisability of our results. The majority of respondents were in their first year of medicine, and therefore their limited exposure to medical training may have influenced their responses. However, there was no differences in the age or gender distribution among responders across the four years or differences in individual TCI trait levels. Certainly this study should be repeated with other medical student cohorts to see if profile patterns are similar. The present data are cross-sectional so they do not demonstrate causal relations or the ability to change personality. However, other experimental research shows that it is possible to help a person grow in self-awareness and thereby change in personality and improve well-being (*Cloninger & Zohar, 2011*; *Krasner et al., 2009*; *Campanella et al., 2014*). Strengths of the study include a large sample of students and the use of well validated measures of our variables.

## CONCLUSIONS

This research identified particular combinations of personality traits associated with low Tolerance of Ambiguity and high levels of maladaptive Perfectionism. These findings have implications for well-being in the medical setting, as they reduce the coping ability of medical students in this challenging profession. The psychobiological model of personality provides insight into traits that are stable and those that are amenable to self-regulation through learning, hence can be targeted in education and training settings. The interaction between biological mechanisms and socio-cultural learning is relevant to a sample of medical students because it accounts for interaction of the biological or innate aspects of their personal development within an intense and competitive learning environment of medical school.

### Funding

Janni Leung is supported by funding from The University of Queensland. The funders had no role in study design, data collection and analysis, decision to publish, or preparation of the manuscript.

### Grant Disclosures

The following grant information was disclosed by the authors:
The University of Queensland.

### Competing Interests

Diann S. Eley and Robert Cloninger are Academic Editors for PeerJ. Kevin M. Cloninger is employed at The Anthropedia Foundation.

### Author Contributions

- Janni Leung and Diann S. Eley conceived and designed the experiments, performed the experiments, analyzed the data, contributed reagents/materials/analysis tools, prepared figures and/or tables, authored or reviewed drafts of the paper, approved the final draft, substantial contributions to the interpretation of data for the work.
- C. Robert Cloninger conceived and designed the experiments, contributed reagents/materials/analysis tools, authored or reviewed drafts of the paper, approved the final draft, substantial contributions to the interpretation of data for the work.
- Barry A. Hong and Kevin M. Cloninger authored or reviewed drafts of the paper, approved the final draft, substantial contributions to the interpretation of data for the work.

### Human Ethics

The following information was supplied relating to ethical approvals (i.e., approving body and any reference numbers):

Ethics was approved by the Behavioural and Social Sciences Ethical Review Committee at The University of Queensland. (Approval Number: 2015001895).

### Data Availability

Data is available as a Supplemental File.

### Supplemental Information

Supplemental information for this article can be found online at http://dx.doi.org/10.7717/peerj.7109#supplemental-information.

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
