# Peer review of "Temperament and character profiles of medical students associated with tolerance of ambiguity and perfectionism"

_PeerJ, doi:10.7717/peerj.7109_

## Round 0.1 · original submission · Major Revisions

The reviewers see merit in the paper, but have some good suggestions for improving its clarity.

·

Basic reporting

Generally the writing was clear and comprehensible. The introduction I found a little confusing as there were frequent lists of facts or measure or components of various personality frameworks. This meant the bigger picture ideas could be a little lost. At times the lists were note directly linked to the research presented (for example, lines 72 – 76, where the various measures used by researchers to examiner the big 5 personality traits were presented, but weren’t really relevant to the study or the interpretation of the study results). I think the authors could streamline the introduction little to allow greater focus on the main points – which in this example would be that although measuring personality using the big five is more common in education research, there are other frameworks that could be used (TCI) which consider personality traits in two categories (temperament traits and character traits).

Experimental design

I wonder if the hypothesis would benefit from being written in the same direction as the analysis – e.g. the analysis focused on determining if personality profile predicted tolerance of ambiguity and perfectionism (both concern over mistakes and high standards), could the hypothesis then have been expressed as expecting students with a personality profile of low harm avoidance and high persistence, self-directedness and cooperativeness to report high tolerance of ambiguity and adaptive perfectionism. This might have made it all link together a little more for the reader.
More information on recruitment would also have been useful to the reader. Whilst the overall response rate was good (62%), participants were predominantly in first year. Did recruitment tactics vary between year groups?
If I’m correct could the same participant have completed the study twice – in year 2014 and again in 2015? If so this could be a potential limitation of the study and needs to be acknowledged as such in the discussion.
It’s not clear why the age groups were chosen as under 25, or 25 or over – what is the rationale for this? This needs to be added to the methods.

Validity of the findings

There is a potential confounder of age and progression through training. The authors briefly mention in the discussion (line 331) that there were ‘no differences in the age or gender distribution among responders across the four years …’ but don’t really address this fully and don’t present the data on this. As there were considerable differences in the numbers of participants between the year groups I wonder what analysis was carried out to determine no difference in age between year groups? Something to clarify this confounder or lack of confounder would be needed.
Alongside this is a more fundamental question. If character traits are amenable to change through social learning and environment, could they have varied with year of study and could this have impacted upon the results of the study? The authors raise this point in the discussion (line 321) when they notes that a longitudinal design would enable them to ‘..examine how personality and coping interacts over time…’ and (line 324 – 325) ‘Further research will follow students’ through medical school to examine changes in personality profiles…’. I think a greater examination of year of study or an acknowledgement that there may be a year of study confounder is needed.
One other point which puzzled me was in relation to the personality profiles. From Figure 1, it would appear that the profiles vary on the levels of harm avoidance (this is in the profile descriptions), in the levels of persistence (mentioned in the discussion - line 294 – 295, but not results), and in self-directedness, which is not mentioned anywhere. Is this the case? If so, was this analysed by the authors? High self-directedness would seem to be important to the authors views regarding potential route of impact of the profiles. Clarity is needed here.

Additional comments

Whilst this was an interesting paper, I think work is required on making it clearer for the reader, both in terms of the methods used, but also in interpreting the results. The lack of clarity on some aspects of the methods would require another review.

·

Basic reporting

The writing is clear and unambiguous, it is of appopriate length and depth
Good coverage of personality literature.

I think there is a gap in the background. The authors talk about some work about tolerance of ambiguity. The article would really benefit from in-depth review and analysis of the clinical reasoning and clinical decision making - as ambiguity and the clinical reasoning process it at the heart of that. It is unclear to me how or if tolerance of ambiguity might relate to clinical decision making or patient outcomes and so situating it the medical education literature would really help to contextualise it.

Experimental design

Research question is defined clearly.

Design is self-report and cross sectional which was appropriate.
More context needs to be provided about the medical students and their training - e.g. most of the sample were first years - was this intended? how much clinical medicine are they doing at the time?

No information is given about response rate or recruitment methods.

I didn't understand the split in under 25 and 25 or over years of age - students in each group could be a month apart in age and the split seems artificial without further statistical justification of this.

Validity of the findings

Data analysis seems robust.

It would be helpful to add in - how representative was the sample of the cohorts?

Were there differences between the years of study((in the discussion I can see that there are no age or gender differences) - medical students in these years are learning and practicing in very different ways and I think results should be presented according to this (rather than age - as students can start their course at different ages) - or statistical justification of why this isn't necessary is presented.

I don't understand the implications of this work - for example, a certain level of tolerance of ambiguity as useful for their clinical practice - even whilst they might find this difficult or stressful. I am also not sure what medical schools could or should do as a result of this link - so not sure that it is a robust conclusion that medical educators should help students at varying levels.

Reviewer 3 ·

Basic reporting

The introduction was clearly structured and methodical however occasionally became slightly confusing. At times the flow of the introduction was lost when accounts were given in relation to different tools utilised within this area 71-86 If structured more methodically in relation to this it is felt the linkage between the different areas of temperament, tolerance and perfectionism would be clearer and would add to the clarity of the study. Terms introduced such as ‘heritable’ 102 would benefit from further explanation within the introduction since it is referred to further within the paper.

The introduction would benefit from more detail relating to medical student experience as they progress through their studies, since different year groups were involved in this present study. Also expansion on what we know from existing literature relating to empathy/burnout/professional behaviour would help give more context to why it is so important to research this area. It is referred to, line 64, but would benefit from more detail and would improve justification for conducting this study.

Overall clearly laid out, methodical structure leading the reader through the background relating to this topic area. The presentation used within this work is clear and allows for effective interpretation of findings and transferability to medical educators worldwide. This work identified clear relevance both for wellbeing of medical students and ability to engage with the requirements placed on them as they progress through their professional career. Provision of raw data, tables and figures gave this work transparency in a varied and well-presented format.

Experimental design

Good piece of research using a unique approach that adds to the existing knowledge base exploring medical student wellbeing. The aims of the paper were clearly presented and understandable (148-157). It is felt that if the introduction was structured more methodically the link with the aims would be clearer. I commend the researchers on exploring this very relevant topic area with its important implications for medical education and managing to collate data from such a large group of participants.
The methods clearly presented the approach to data collection of validated tools, and acknowledge ethical approval. Some more detail regarding sample would be helpful such as were they just undergraduate medical students or were they direct entry 168.
Appropriate account of approach to analysis of data collected. Clear description of tools implemented to gather data, and provision of additional information in appendices was helpful.

Validity of the findings

Unique, transparent approach to a very topical area of medical student wellbeing. Work acknowledges study limitations, 312, 317, 329 and is clear that findings can only suggest conclusions that can be drawn. Identifies important areas of future research within this area; longitudinal/across years. Conclusions well stated and linked to research question providing important information for the area of medical education.

Additional comments

Found this a really engaging piece of work, which contributes to a knowledge base that has important implications for long-term wellbeing and professional implications for medical students.

---

## Round 0.2 · accepted · Accept

Thank you for making the revisions suggested by the reviewers.